# Additive Manufacturing of Prostheses Using Forest-Based Composites

**DOI:** 10.3390/bioengineering7030103

**Published:** 2020-09-01

**Authors:** Erik Stenvall, Göran Flodberg, Henrik Pettersson, Kennet Hellberg, Liselotte Hermansson, Martin Wallin, Li Yang

**Affiliations:** 1Stora Enso AB, Sommargatan 101A, 65009 Karlstad, Sweden; erik.stenvall@storaenso.com; 2RISE—Research Institutes of Sweden, Drottning Kristinas väg 61, 11486 Stockholm, Sweden; goran.flodberg@ri.se (G.F.); henrik.pettersson@ri.se (H.P.); 3Embreis AB, Tumstocksvägen 11 A, 18766 Täby, Sweden; kennet.hellberg@embreis.com; 4Department of Prosthetics and Orthotics, Faculty of Medicine and Health, Örebro University, 70185 Örebro, Sweden or liselotte.hermansson@oru.se (L.H.); martin.wallin@regionorebrolan.se (M.W.); 5University Health Care Research Center, Faculty of Medicine and Health, Örebro University, 70185 Örebro, Sweden

**Keywords:** biocomposite, forest-based MFC, fibrils, additive manufacturing, artificial limb, fused deposition modeling (FDM)

## Abstract

A custom-made prosthetic product is unique for each patient. Fossil-based thermoplastics are the dominant raw materials in both prosthetic and industrial applications; there is a general demand for reducing their use and replacing them with renewable, biobased materials. A transtibial prosthesis sets strict demands on mechanical strength, durability, reliability, etc., which depend on the biocomposite used and also the additive manufacturing (AM) process. The aim of this project was to develop systematic solutions for prosthetic products and services by combining biocomposites using forestry-based derivatives with AM techniques. Composite materials made of polypropylene (PP) reinforced with microfibrillated cellulose (MFC) were developed. The MFC contents (20, 30 and 40 wt%) were uniformly dispersed in the polymer PP matrix, and the MFC addition significantly enhanced the mechanical performance of the materials. With 30 wt% MFC, the tensile strength and Young´s modulus was about twice that of the PP when injection molding was performed. The composite material was successfully applied with an AM process, i.e., fused deposition modeling (FDM), and a transtibial prosthesis was created based on the end-user’s data. A clinical trial of the prosthesis was conducted with successful outcomes in terms of wearing experience, appearance (color), and acceptance towards the materials and the technique. Given the layer-by-layer nature of AM processes, structural and process optimizations are needed to maximize the reinforcement effects of MFC to eliminate variations in the binding area between adjacent layers and to improve the adhesion between layers.

## 1. Introduction

Fossil-based thermoplastics are the dominant raw materials in both prosthetic and industrial applications; there is a general demand for reducing their use and replacing them with renewable biobased materials [1,2]. Forests hold the biggest share of renewable biomaterials on earth, which are inevitably central to the biobased economy. Various biocomposite materials have been studied and also commercialized to some extent in recent years. Biocomposites, also called natural fiber composites, can be made from many different types of plant fibers originating from bast, leaf, straw, grass, or wood [3]. Natural fibers from wood are normally pretreated by grinding and after this they are treated again by mechanical or chemical means to release fibrils from the wood fiber. Instead of having only fibrils of a narrow size distribution, microfibrillated cellulose (MFC) can contain fibers, fiber fragments, fibrillar fines, and nanofibrils [4,5]. Then, the MFC is melt blended with thermoplastic polymers in an extrusion process to form biocomposites. MFC can significantly improve the strength and stiffness when used as reinforcement in thermoplastics [6]. In additive manufacturing (AM), industry fiber reinforcement is often needed to meet the strength requirement or other functions [7,8,9] to replace traditional materials such as aluminum in many applications [10]; in this context, MFC becomes an interesting renewable-source reinforcement.

Prosthetic and orthotic products are important for healthcare and well-being. Today, approximately 3000 prosthetic limbs are manufactured by different companies in Sweden, with an estimated value of 100 million Swedish kronor (MSEK) per year. The materials used are fossil-based thermoplastics and reinforcement materials, for example, glass or carbon fibers, metals, etc. According to a market study, there are more than 100 million individuals worldwide with limb loss and more than one million amputations annually. In addition to four million people living with limb loss in 2014, there are 400,000 amputations/year in the USA and EU countries. The global prosthetics and orthotics market size has been estimated to be USD 9.2 billion in 2019 with an annual growth rate of 4.6% [11].

A prosthetic or orthotic product is unique and custom made for each patient. Specialists make the customization to ensure a good result for the patient. It requires multiple fitting-and-adjusting cycles, which is both time consuming and expensive. The specialist uses many different types of materials for the customization, for instance, thermoplastics, such as PP, PE (polyethylene), PETG (polyethylene terephthalate glycol), and PA (polyamide), as they are easy to customize; carbon fiber composites and special high-strength titanium alloys and aluminum alloys are also commonly used to ensure that the prostheses are both lightweight and durable, and soft foam and gel materials from silicone, EVA (ethylene-vinyl acetate), PUR (polyurethane), PE and SEBS (styrene ethylene butylene styrene) are used to provide cushioning effects.

Additive manufacturing (AM), also known as three-dimensional (3D) printing, is rapidly progressing in industrial application prototyping. Unlike traditional manufacturing processes, in which tools are needed, and geometrical complexity and production volume are often limiting factors, AM needs no tools, hence, it is not limited by either geometrical complexity or production volume, which makes it a perfect technique for personalization from design and prototyping to production. Due to a strong interest worldwide, many AM techniques have been developed in recent decades. These techniques operate on different principles, from binding, fusing, and melting to polymerization, and with different materials including sand, polymer, metal, and biomaterials [12]. Nevertheless, the number of polymeric materials available for AM applications based on FDM (fused deposition modeling) and polymer bed fusion (scanning laser sintering (SLS) and digital light processing (DLP)) are rather limited, which prevents the adoption of the techniques. The qualities of FDM-produced products are affected by various process parameters, for example, layer thickness, build orientation, raster width, and print speed. The process parameter settings and their ranges depend on the type of FDM machine. The optimum parameters can improve the qualities of three-dimensional (3D) printed parts [13].

In the last few decades, AM techniques have been introduced into the prosthesis sector [14,15] and AM is now considered to be an emerging prosthetic production technology. The number of businesses that offer prostheses produced by 3D CAD (additive manufacturing) is increasing rapidly. The first prosthetic parts that were additively manufactured were cosmetic outer shells for lower limb prostheses, in which the AM technique made a perceptual impact on the prosthetic and orthotic products. From the first production of cosmetic shelves, this technique is now suggested, for example, for production of prosthetic sockets [16] and prosthetic hands [17]. However, there is a scarcity in the literature regarding 3D-printed prostheses for the lower limbs. A recent scoping review [18] only found 11 articles that covered the implications of 3D printing in the field of lower limb amputation, but they concluded that it was a promising advancement in modern prosthetic fabrication. Among the existing studies, the focus has mainly been on user experiences and improved functionality of the AM-fabricated prosthetic parts using commercial materials [19,20,21].

The transition from conventional manufacturing to 3D printing of entire prostheses is not easy. A unique prosthesis must be created in a digital format, and then produced through AM [22]. Moreover, the AM materials must be high strength, light weight, and produced at a reasonable cost. In their 2020 review, Barrios et al. concluded that one of the future paths in this field would be based on the design of new materials [14]. However, the additively manufactured products, for example, lower limb prostheses, must be able to tolerate high and dynamic loads and provide sufficient durability. The mechanical properties of prosthetic sockets produced with this technology have yet to be proven. The strength and durability of the sockets were identified as important aspects as early as 2005 [16]; a review in 2018 [23] further emphasized the need for quality control and testing of the sockets. However, to our knowledge, no lower limb prosthesis, socket or other product component, has been produced using renewable source MFC-reinforced materials and AM technologies.

Our hypothesis is that MFC from wood fiber for reinforcement of composites combined with AM techniques enables innovative and personalized prosthetic solutions. The focus of this work is proof of the concept, i.e., obtaining MFC-based composites that are appropriate for FDM and prosthetic applications. The results presented were generated in two research projects. The aim of the projects was to develop systematic solutions to combine biocomposites with AM techniques, for example, FDM and SLS, through close collaboration between partners along the value chain from material development to end-user applications. Prosthetic products were chosen because they offer excellent opportunities for developing both AM and biocomposites, as there are well-established testing methods and quality requirements and standards which are often higher than for ordinary consumable products. Hence, prosthetic products are well suited to our purpose of technical developments in biocomposites for 3D printing applications. Prior to this work, there have been two project publications. Pore characteristics of SLS-built parts of nylon 12 with or without the addition of carbon fibers were found to be responsible for the variation of the parts’ mechanical performance [24]. It has also been demonstrated that topology optimization based on FEM (finite element modeling) could be used in prosthesis design and structural optimization [25].

## 2. Materials and Methods

This section describes the composite materials, the processing techniques for the test specimen and demonstrators, the mechanical and structural test and measurement techniques, and a survey of the end user’s attitude towards biobased composites and 3D-printed prostheses.

### 2.1. Composite Materials

Extensive studies on material composition and characterizations have been carried out. First, multiple trials with different MFC and PP blends were undertaken at a lab scale. Then, two pilot-scale trials were conducted to further verify the results. Finally, the composite materials studied in this work were produced by Stora Enso in a dry compounding process. The composites, called DuraSense PP AM quality, were specially fabricated for this work and consisted of a polypropylene (PP) matrix with 20, 30, and 40 percent MFC made from chemical pulp fiber. It is worthwhile noting that the quality of the MFC used in this work was adopted to composite fabrication. According to Chinga-Carrasco [4], an MFC could contain several material components, for example, fibers, fiber fragments, fibrillar fines, and nanofibrils. Due to the fact that the composites having different MFC contents were produced on different dates and in different batches, there were possible variations in both materials and production conditions. To improve both the compatibility and the dispersion of MFC in the PP matrix, the following two main strategies were followed: (I) An adapted compounding process at Stora Enso, which facilitates the dispersion of MFC in the specific polymer matrix and (II) the use of a functional coupling agent, which enhances the compatibility between the hydrophobic polymer phase and the hydrophilic MFC. The composite materials were in granulate form and suitable for FDM-based AM applications.

### 2.2. Production of the Test Samples and the Demonstrators

The ISO 3167 multipurpose test samples (dumbbells) were produced on a BOY 25 EVH injection molding machine from Dr. BOY GmbH & Co. KG, Neustadt-Fernthal, Germany, using the parameters listed in Table 1. Similar test samples were created with the FDM technique. The FDM machine was an ErectorBot 644LX (ErectorBot, Inc. CA), equipped with a custom extruder printhead with a nozzle of diameter 0.8 mm. The test specimens were built with either longitudinal lines or perpendicular lines inside a perimeter frame, as shown in Figure 1. The other parameters applied were line width of 0.9 mm, layer height of 0.3 mm, print temperature of 240 °C, and print speed of 33 mm/s.

Figure 2a shows a transtibial (lower limb) prosthesis, which was the demonstrator for this work created using the FDM technique with biocomposite materials. To demonstrate AM’s geometric precision, strength, and durability capabilities, a test socket (Figure 2b) was also 3D printed with similar, however, simpler geometry than the lower limb prosthesis. The socket was regarded as a building block of the lower limb prosthesis, and the simplified geometry enabled testing according to the ISO standards.

### 2.3. Mechanical Testing, SEM and X-ray Microtomography

The static ultimate test and the dynamic test were performed at Fillauer Europe AB and Embreis AB according to ISO 10328:2016 [26], a structural standard test for prosthetics regarding the structural testing of lower limb prostheses. Loading condition II was applied with P6 test geometry, which meant that the tested sockets experienced not only off-axis compression and bending but also twisting around a few axes, as illustrated in Figure 3a. The test equipment shown in Figure 3b include a Static Test Machine No. K1 Tinius Olsen 25ST from Tinius Olsen TMC, USA and a Sauter TVS 10KN100 from SAUTER GmbH, Balingen, Germany. The load was measured using a 10 kN load cell with a measurement accuracy of +/−5 N.

Electron scanning microscopy (SEM) was carried out using a Hitachi SU3500, Tokyo, Japan. The instrument is equipped with a backscattered electron detector (BSE), secondary electron detector (SE), and X-ray energy dispersive spectrometer (EDS). The pictures shown in this work were taken with the BSE detector at 13 kV and at 250X magnification.

In recent years, X-ray microtomography has become a valuable tool for material characterization [27]. The X-ray microtomography was performed with an Xradia MicroXCT-200 (Carl Zeiss X-ray Microscopy, Inc, Pleasanton, CA, USA). The scanning conditions were as follows: X-ray source (voltage 40 kV, power 4 W), the number of projections was 1.289, and the exposure time was 20 s/projection. The distances from the detector and from the X-ray source to the sample holder were 16 and 30 mm, respectively. The magnification was 20×, the pixel size of the image was 0.8149 µm, the pixel resolution was 1.07 µm, and the maximum analyzed volume was 1 × 1 × 1 mm. The samples were examined in their X, Y, and Z directions.

The injection molded specimens for both tensile and impact testing were kept conditioned at 23 °C, 50% RH, for a minimum of 48 hours prior to testing. The tensile tests were performed using a hydraulic actuator with an MTS FlexTest™ 60 digital controller (MTS Systems Corporation, Eden Prairie, MN, USA). The specimens were fastened using clamps, and a contact extensometer was attached to the specimens with rubber bands. The load was measured with a 40 kN load cell, and the tests were performed at a rate of 50 mm/min. Data were recorded at a sampling rate of 500 Hz using MTS Series 793 control software. The impact tests were performed on a Zwick HIT5P pendulum impact tester (ZwickRoell GmbH & Co. KG, Ulm, Germany) equipped with a 5 Joule Charpy pendulum.

The dynamic viscosity of the composite material was measured using a Malvern Rosand RH 10 capillary rheometer with 15 mm twin bore at 190 °C.

### 2.4. User Attitude

Initially, a study-specific survey was constructed to inquire about the users’ attitude for transferring production of orthotic and prosthetic devices to a more sustainable material. The users were asked how likely it was that they would choose a product produced with a biocomposite rather than fossil-based thermoplastics if it meant a compromise in design, color, or quality. The survey also asked about potential benefits or risks of using the new material, and what products the users would like to see made from this material. Outpatients visiting the department of Prosthetics and Orthotic, Örebro University Hospital, Örebro, Sweden, answered the waiting-room survey. The demonstrator was tested by a patient with a transtibial (lower limb, below-the-knee) prosthesis. The patient was 70 years old, a female with 52 years of experience using a prosthetic limb every day. The outcome was reported by the Satisfaction with Prosthesis subscales of the Trinity Amputation and Prosthesis Experience Scales, Revised (TAPES-R) [28,29]. These subscales have 3 (aesthetic) and 5 (functional) subscale items, scored on a 3-level rating scale in which a high score is positive, indicating satisfaction with the prosthesis. The patient responded to the questionnaire before and after wearing the demonstrator for a comparison with the conventional prosthesis.

## 3. Results

### 3.1. Mechanical Properties of the Biocomposites

To evaluate the characteristics of the MFC composite materials to be used for FDM, standard test specimens were injection molded, followed by tensile testing. The distribution of cellulose material in the PP matrix was examined by microtomography. Table 2 and Figure 4 show that the tensile strength (σ_m_) and the tensile moduli (E_t_) were significantly improved by the addition of MFC to the composites. Six samples were tested for each data point. Figure 4 shows that the ultimate tensile strength increases significantly with increasing MFC content. The tensile strength is about twice as much with 30 wt% MFC as compared with neat PP. The improvement in tensile strength becomes less significant with additional increases in MFC content. This could be because a higher fiber content resulted in more fiber damage and shorter fiber lengths. Fu and colleagues [30] showed that the mean glass and carbon fiber lengths in their PP composites after injection molding decreased with increasing fiber volume fractions. As a result, the reduction in mean fiber length offset the reinforcement effect by increasing fiber volume fraction. In addition, the E_t_ modulus values of the composites monotonically increased with MFC content. This was also observed by Fu et al. [30], for example, the modulus for both types of composites increased dramatically with increased fiber volume fraction. This indicated that the composite modulus depended on MFC volume fraction rather than length. Nevertheless, one should bear in mind that these composites, in Table 2, were produced in different batches with different raw materials and processing settings. Therefore, the differences in their mechanical properties are a collective reflection of several parameters rather than purely MFC content, and the comparison thus becomes indicative.

The elongation at break (tensile strain, ε_b_) of the composites decreases with increasing MFC content, indicating more brittle-like fracture of the biocomposites as compared with neat PP. The impact strength was measured using the Charpy notch test method in which a notch was made in each test sample. Table 2 and Figure 5 show that the Charpy impact strength decreases with the addition of MFC, which is often the case due to reduced elasticity and flexibility with increased reinforcement as compared with neat PP. Similar observations have been reported by several authors [31]. The explanations were that the fibers induced a fracture change from ductile to brittle and increase fiber agglomeration, which also resulted in high-stress regions and non-uniform stress transfer.

The aspect ratio (length/diameter) of wood pulp fibers usually ranges between 44 and 75, depending on the processing method [32], whereas the aspect ratio of MFC can be up to several hundred due to its small diameter (<100 nm) and long fibril length (µm), depending on the fabrication method [33,34]. The higher the aspect ratio, the higher the reinforcement capacity when incorporated in composite materials [35]. The MFC, however, can consist of several inhomogeneous material components, for example, fibers, fiber fragments, fines, and fibrils [4]. A detailed analysis of the fiber dispersion is given in the next subsection.

Standard composite test specimens consisting of 20 wt% MFC were also created with the FDM technique. The mechanical properties of these test specimens are shown in Table 3. Three samples were tested for each data point. The tensile strength and modulus of the FDM test specimens built with longitudinal lines are approximately 68% of the injection molded values shown in Table 2. It was reported by Lay et al. [36] that the mechanical strengths of the FDM-built parts of three polymers, i.e., PLA (Polylactic acid), ABS (Acrylonitrile butadiene styrene), and nylon 6, were 48, 34 and 37% lower than their counterparts fabricated with injection moulding. In our case, the mechanical strength of the FDM built parts is about 32% lower than the injection moulding.

It is well known that adding fibers in PP significantly increases the shear viscosity, and thus influences the processing behavior (flowability) of material deposition during the FDM process. Figure 6 shows that the biocomposites made of PP/MFC have a strong shear thinning behavior, resulting in a much lower viscosity and resistance to flow at high shear rates. It was also observed in this study that the processing temperature had a significant influence on the dynamic viscosity, which was similar to most thermoplastics.

### 3.2. Fiber Dispersion and Pull-out for the Biocomposites

The uniform dispersion of MFC additives is crucial to the mechanical strength of composite materials. As stated in Section 2.1, the following two main strategies were followed to improve both the compatibility and the dispersion of MFC in the PP matrix: (I) An adapted compounding process which facilitates the dispersion of MFC material components in the polymer matrix and II) the use of a functional coupling agent, which enhances the compatibility between the hydrophobic polymer phase and the hydrophilic MFC components.

Figure 7 and Figure 8 show that the cellulose material is uniformly dispersed in the PP matrix. However, single MFC fibrils are not visible in these images due to limited resolution. Much of the reinforcement effect observed in the tensile strength and Young’s modulus can be attributed to MFC with a large L/D ratio. Another important factor is that a larger fiber specific surface area, such as in MFC, promotes binding of the fibers with the polymer matrices, resulting in improved mechanical strength.

Figure 7 shows that no cellulose material agglomerates are visible in the x-y cross-sections, even though they often occur in biocomposites due to strong fiber–fiber respective fibril–fibril interactions and the hydrophilic character of the MFC components, in contrast to the hydrophobic nature of polypropylene. Agglomeration results in regions where less energy is required to begin a crack [31]. 

Figure 7 also shows black spots or holes in the cross-sections in the images, which most likely originate from moisture. MFC holds water strongly, and it is difficult to obtain a completely dried cellulose material in the PP matrix, although it might be achievable in up-scaled production facilities. These holes occur randomly for all MFC composites and are visible in microtomography in both the x-y and y-z planes. They can affect the mechanical properties because they can form cracks.

The fiber-matrix interface property also affects the composite’s mechanical behavior, of which the adhesion between fiber and polymer matrix is the key [37]. One common way to observe the interface effect is through a fiber pull-out investigation, which consists of a debonding process followed by a pull-out process. Since MFC contains fibers and fiber fragments, therefore, pull-out testing is possible. In this work, samples from a Charpy notch test were investigated, using an SEM technique, shown in Figure 8. As shown in the images, only a few sites could be connected to either fiber pull-out or gas formation in processing. The latter originated from the low water content in the hydrophilic fibers, as was also observed in the X-ray microtomographic images in Figure 7. Because these sites are few and well-separated from one another, they are not expected to be the major cause for the reduction in impact strength. In fact, impact strength between the biocomposites and the neat PP is difficult to compare since the materials have such different ductilities. The neat PP is very ductile and it is difficult for a crack to propagate, while the biocomposites are much stronger and much more brittle, allowing cracks to easily propagate inside the material. The crack propagation characteristics can, to some extent, be circumvented by using impact modifiers according to Thomason et al. [31], but this is beyond the scope of this work.

### 3.3. Mechanical Performance of the Test Socket

Polypropylene (PP) is a commonly used material for prosthetists and orthotists. Table 2 shows the test results of the new material with polypropylene (PP) as a matrix and 20% MFC made from chemical pulp fiber produced a tensile strength of 32.10 ± 0.15 MPa, more than 30% higher than that of PP. This material was used for additive manufacturing based on the FDM process.

Figure 9 shows the test result (loading history) of the FDM-manufactured socket with the biocomposite containing 20% MFC. The socket was broken at 2896 N, equivalent to 12.9 MPa. The test socket’s mechanical performance was evaluated according to ISO 10328:2016. Loading condition II was applied using P6 test geometry, as shown in Figure 3a. This means that the socket was simultaneously subjected to compression and off-axis bending and also torsion along several axes, which simulates the worst-possible load situation during normal use of a prosthetic lower limb. Hence, the strength obtained from this test is not directly comparable with the tensile strength listed in Table 2 and Table 3.

Three sockets were tested at the end of this work. These sockets were manufactured using the FDM technique with the same composite material (PP/MFC 80:20) but in different build orientations, i.e., the transverse plane, the coronal (frontal) plane, and the sagittal plane. The test results are listed in Table 4, wherein “transverse plane” indicates the deposition layer-by-layer is in the transverse plane and so forth. The force at break (the strength of the test socket) is highly dependent on the build orientation. The strength of the socket built in the sagittal plane is twice that of the transverse plane. The underlying explanation for why the strength of the sockets built in the coronal plane and the sagittal plane was much higher than that of the sockets built in the transverse plane is that when printed in these directions, the MFC reinforcement was better realized. This reveals tremendous FDM process impacts on the final mechanical performance. The strength would be even higher if we could improve adhesion between the layers.

Every test socket broke at the distal end where there was an internally sharp corner, and the load was the highest according to FEM simulation [25]. The sharp corner is a starting point for the fracture failure and the specific load application produces axial compression, shear forces, bending moments, and torque as a result of the load vector in load condition II. Therefore, the strength (MPa), listed in Table 4, is a combined strength caused by the force and the load geometry. The load geometry and the force that a prosthetic limb shall sustain according to ISO 10328:2016 comes from real-life measurements on prosthetic lower limb users. This makes it impossible to directly compare the strength of the material in the socket form (Table 4) with the tensile strengths obtained from the injection molded test samples (Table 2).

### 3.4. Analysis of Consumers’ Attitude towards Biocomposites

The consumers (n = 27) reported that they most likely would choose a product made from biocomposite over a conventional plastic product unless it meant having to compromise quality (Figure 10a). The acceptance towards biocomposites was not affected by compromising in design or color, only 8% of the respondents were likely to avoid these products (Figure 10b). They reported no or low fear of risks related to the new material, and instead were interested in the environmental advantages such as biodegradability and recycling. When asked what products the consumers would like to see made from this new material, they responded "all parts that today are made from plastics”. Example suggestions included all sorts of orthoses, i.e., insoles, corsets, and breast prostheses. The reasons stated included the environmental benefits, and also that these products came in close contact with the skin and were used quite extensively.

### 3.5. Patient and Clinician Experiences with the 3D-Printed Prosthesis 

Preparation of the product prior to client fitting showed that the biocomposite material was easy to cut with a jigsaw and easy to smooth the surface by machine grinding. Spontaneous comments from technicians were that working with the material in preparation for the socket created a pleasant wood smell. The socket demonstrated sufficient form stability to provide the correct pressure distribution in the socket and enough overall stability to make the prosthesis useable for gait in a lightweight female.

The patient reported that wearing the demonstrator felt no different from wearing the conventional prosthesis (Figure 11a). However, according to the patient, the demonstrator had a better color; and she would have chosen this demonstrator over a conventional prothesis had she had the opportunity to choose. This was reflected by the patient’s score on the TAPES-R, where the overall score on the aesthetic satisfaction subscale was improved from six to seven; the satisfaction with appearance item score increased from two = satisfied to three = very satisfied; whereas on the functional satisfaction subscale the overall score was the same (10 = satisfied) before and after wearing the demonstrator. Moreover, despite the actual increase in weight (the conventional laminated socket without adaptors weighs 148 grams versus 393 grams for the experimental socket (the demonstrator)), the patient reported a sense of decreased weight of the demonstrator as compared with the conventional prosthesis during walking (Figure 11b).

## 4. Discussion

From a material point of view, biocomposites offer a renewable and sustainable material alternative for AM-processed applications to traditionally used thermoplastic or thermosetting materials. However, biocomposites are new materials for this type of processing and there is still much to understand and develop in regard to MFC dispersion in the polymer matrix, MFC aspect ratio, MFC size and orientation, and interfacial strength between the MFC and the polymer, etc. Although the MFC materials clearly reinforce the polymer matrix, such as PP, it is very hard to transfer this reinforcement potential between layers in 3D printing. Thus, it is very important to choose the build orientation to maximize the effect of MFC reinforcement, as shown in Table 4. For the same reason, 3D printing processing conditions are very important for the dynamic behavior of the material in the FDM process, layer-layer adhesion, and the final properties of the built parts. In this study, PP was selected as the polymer matrix because it has not commonly been used in 3D printing but has been used extensively in the injection molding industry due to its beneficial mechanical properties in relation to its cost. It was observed that the MFC improved the melt-flow properties of PP composite and therefore enabled its use in 3D-printing. In the future, PP from renewable sources, which were not available when the project was started, could be used to further strengthen the sustainability aspect.

The overall mechanical strength of the additively manufactured object (AMO) depends on the composite material and also on the structural characteristics of the printed objects. The former determines the upper limit of the mechanical strength achievable with this material, whereas the latter determines the actual strength achieved with the AM technique. The overall mechanical strength of the AMO is defined by the weakest link in its structure. Hence, it is particularly important to identify the limiting factors related to the AM techniques.

There are several factors that can affect the overall mechanical strength of AMOs [13]. Voids and porosity in the bulk structure are known to be responsible for the variation in the mechanical strength of SLS processes [24]. They also exist in structures created by FDM, as the top and bottom filaments typically do not attach perfectly and bond to each other, forming air pockets and porous structures with large gaps between the strands [36]. Figure 12 clearly shows the voids at the start and end positions of each layer and even between the layers. Voids and pores are of particular concern to the mechanical strength when their locations are at or close to the AMO’s surface. Variation in the effective binding area between adjacent material layers can also cause deteriorated mechanical performance. The area variation could have resulted from the dimension accuracy related to the precision of the print-head’s movement in FDM, morphology of the underlying layer, the temperature gradient between the layers, etc. Moreover, there could have been a systematic area reduction due to the spatial gradient of the 3D model, as illustrated in Figure 13 (marked by red rings). This type of reduction is particularly severe with a thick building layer. Therefore, the overall mechanical strength can be improved using hardware and software improvements to reduce or even eliminate random and systematic binding-area variations. In addition, the mechanical strength of AMO can be further improved by testing different slicing, temperature, or printing speed settings, or in different ambient temperatures, etc. [13].

Additive manufacturing can potentially enable paradigm changes in the prosthetic value chain from design to terminal devices. The changes benefit patients as follows: They can receive their prosthesis faster; the digital prosthesis is easy to store and retrieve, enabling production of multiple devices from the same scan; and the digitalized prosthesis is easy to visualize, therefore, a patient can see what the prosthesis should look like prior to manufacturing. These lead to a more personal solution in which the patient’s concerns and priorities can be addressed. There are many other benefits. With AM, material waste is reduced to a minimum, which together with biobased composites, reduces the imprint of the prosthesis on the environment. The consumers’ view of the environmental aspects of production in this study showed that this is something that the market should consider.

Another important aspect of prosthetics is cost. The cost limits access to prosthetic devices in many parts of the world [38]. Costs relate to both production of the customized sockets and the finalizing of the product using prefabricated devices such as joints, feet, and hands. Very little attention has been given to research on these aspects of prosthetics. Biddiss et al. [38] discussed the implications of modular designs and rapid prototyping along with computer-aided design and manufacturing on the cost of prosthetic components. A review of 3D-printed upper-limb prostheses supported this suggestion [15], showing that the maximum material cost was $500. Another aspect of prosthetics, related to cost but rarely studied, is the implication for the patient’s quality of life. There is a need for more high-quality research that reflects the effectiveness of different prosthesis interventions in terms of users’ quality of life [39].

Although the perspective of combining MFC-reinforced composite with AM is promising for prosthetic applications, there are also challenges. From the prosthetic application point of view, the material looks very promising regarding the way it failed, however, it is unclear whether it is the polypropylene (PP) as matrix, the 20% MFC content, or both together that produces a more ductile break. Due to the layer-by-layer nature, MFC in the composites are less likely to over-bridge different layers. It is very important that the adhesion between each layer is optimal. To maximize the reinforcement effects and to obtain the best material tensile strength, process direction is of crucial importance. This is particularly true for FDM processes. Potentially, the stiffness and durability of a prosthetic socket made from this material could be improved by varying the thickness of the socket and making, for example, a honeycomb structure, especially in the distal parts of the socket. This is one of the advantages of the 3D printing technique. Future development should improve the adaptors to optimize transitioning into and out of the socket. Moreover, cost and education are required to introduce this procedure into industrial applications, both regarding the software and the hardware. Manufacturing with AM requires 3D scanning of the patient, data modulation of the 3D scan, designing the prosthesis in 3D CAD, additive manufacturing, mounting, adjusting, and finalizing the prosthesis.

There are also potential limitations in this study. The initial test of the demonstrator produced in this project showed promising results. However, the product was tested with only one type of suspension, the distal lock-pin attachment, and other systems could function differently with this material and production technique. The sockets produced with this new material and by FDM may not be impermeable to air. Therefore, future testing is required with other types of suspensions, for example, the sleeve-and-valve vacuum system or sealed-in suspension systems. In addition to the dynamic testing, cyclic mechanical testing is also important for prosthetic applications, which need to be complemented in the future. Moisture susceptibility, water absorption, as well as friction and wear behavior [40] could potentially be important aspects which have not yet been examined. Due to the hydrophilic nature of MFC, the composite can take up moisture in a humid environment and make the composite susceptible to microbial growth [41,42]. Espert et al. reported that when immersed in water, the composite’s water absorption followed Fick’s law and the mechanical properties were severely affected [43]. Modifying the cellulosic material to make them hydrophobic or using a cap layer are possible solutions [43].

## 5. Conclusions

Composite materials made of polypropylene reinforced by MFC were investigated. MFC materials (20, 30 and 40 wt%) were uniformly dispersed in the polymer PP matrix, and the mechanical performance of the materials was significantly enhanced by the addition of MFC. The ultimate tensile strength of the composite was about twice that of the PP when the MFC content was 30 wt%, while its Young’s modulus more than doubled. The composite material with 20 wt% of MFC was successfully applied in an AM process using an FDM-based technique; a transtibial prosthesis was created based on the end-user’s data. A clinic trial of the prosthesis was conducted with successful outcomes for wearing (walking) experiences, appearance (color), and acceptance of the materials and the technique. The 30 and 40% MFC composites have higher viscosity, therefore, their application requires a more powerful extruder, beyond the limit of the FDM machine used in this work.

The AMOs created with the FDM technique and the MFC-reinforced material exhibited strong dependence on build orientation. To utilize the full range of mechanical strength offered by the materials, structural and process optimizations are needed to transfer the strength of the material to the AMO’s strength. Due to the layer-by-layer nature, MFC material components in the composites are less likely to over-bridge different layers, which leads to deteriorated tensile strength in the cross direction. Structural and process arrangements must be adapted to the terminal application to maximize the reinforcement effects of MFC, to eliminate variations in binding area between adjacent layers, and to improve adhesion between layers to create robust, durable prostheses using AM techniques.

This study finds that combining biocomposites with 3D printing offers a promising future for prostheses and orthosis solutions. However, further developments in both material and AM technology are needed to enable the industrial sector to achieve sustainability improvements as a result of using renewable-sourced 3D-printed materials.

## Figures and Tables

**Figure 1 bioengineering-07-00103-f001:**
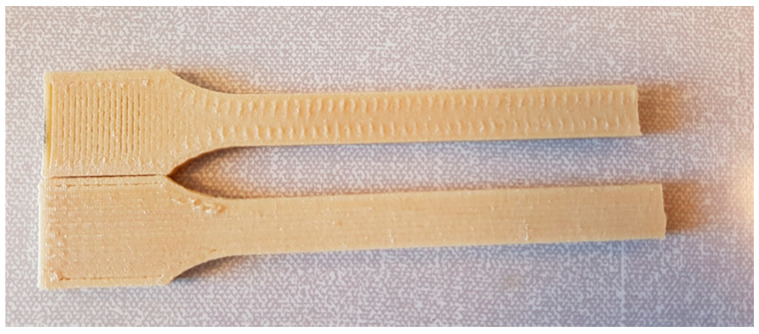
Standard test specimens built with fused deposition modeling (FDM) and with longitudinal lines (bottom) and perpendicular lines (top).

**Figure 2 bioengineering-07-00103-f002:**
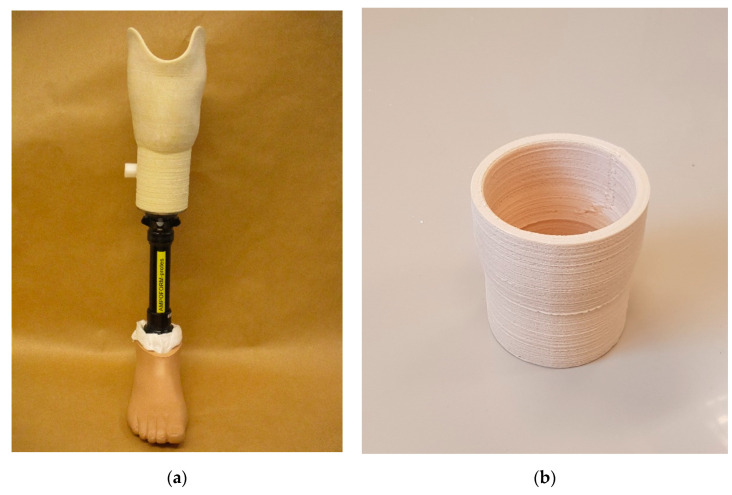
The 3D-printed demonstrators. (**a**) The lower limb prosthesis; (**b**) The test socket, a simplified variant of the prosthetic socket.

**Figure 3 bioengineering-07-00103-f003:**
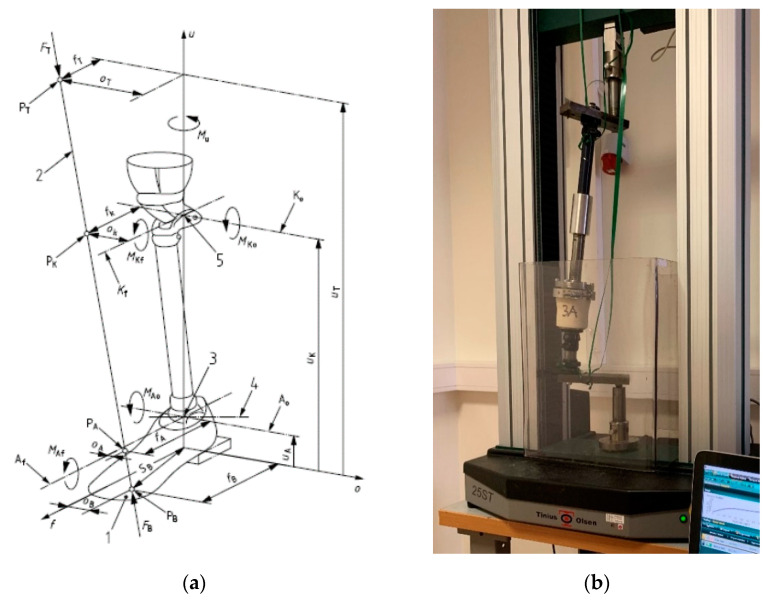
Structural testing method for lower limb prostheses (ISO 10328:2016). (**a**) Illustration of loading condition II; (**b**) Test equipment with the test piece loaded.

**Figure 4 bioengineering-07-00103-f004:**
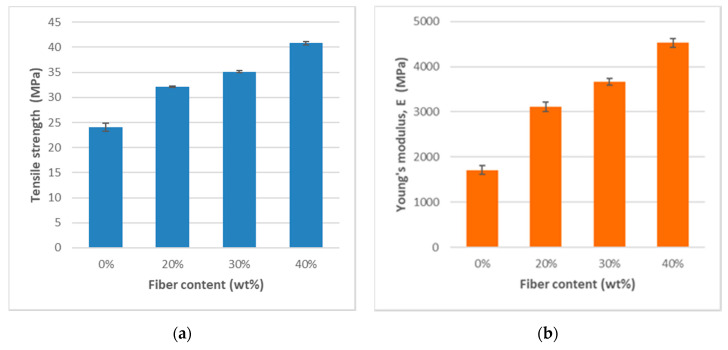
Relationships of the composites’ mechanical properties by microfibrillated cellulose (MFC) content. (**a**) Ultimate tensile strength; (**b**) Tensile (Young’s) modulus.

**Figure 5 bioengineering-07-00103-f005:**
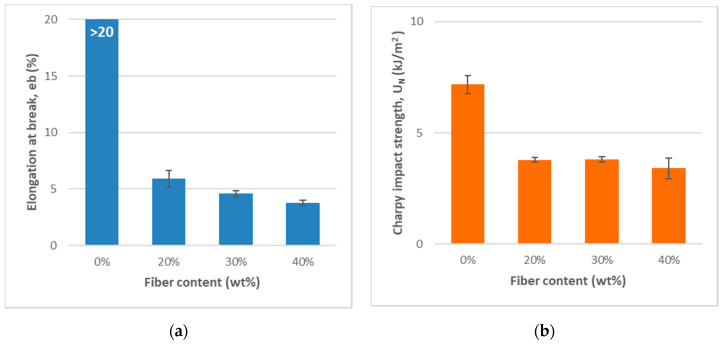
Relationships of the composites’ mechanical properties by MFC content. (**a**)Elongation at break (%); (**b**) Charpy impact strength (kJ/m2).

**Figure 6 bioengineering-07-00103-f006:**
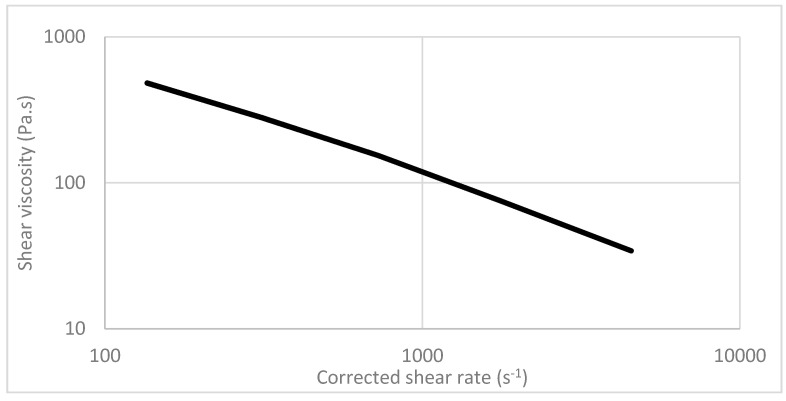
The dynamic shear viscosity versus corrected shear rate for the biocomposite polypropylene (PP)/MFC 70:30 measured with a capillary rheometer at 190 °C.

**Figure 7 bioengineering-07-00103-f007:**
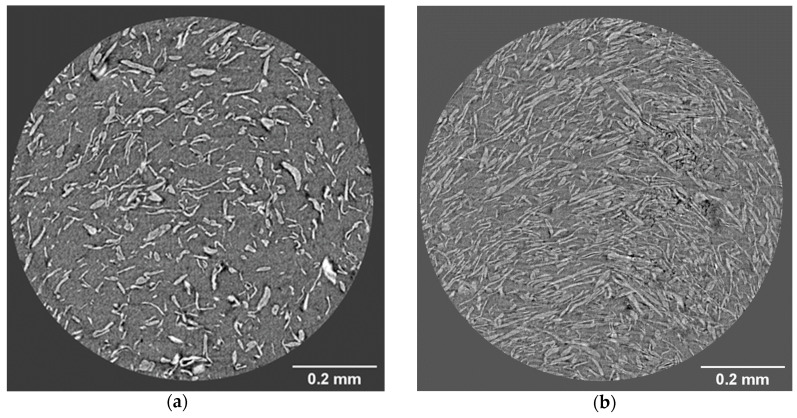
X-ray microtomographic images of the composite cross-section (x-y plane). (**a**) PP/MFC 80:20; (**b**) PP/MFC 60:40. The magnification is 20×.

**Figure 8 bioengineering-07-00103-f008:**
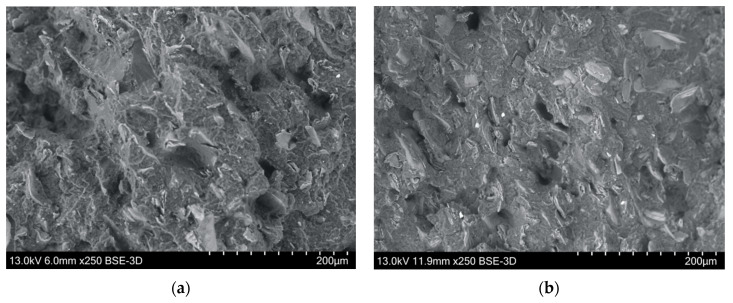
SEM images to identify fiber pull-out in cross-sections of injection-molded test specimens after the Charpy notch test of the composites. (**a**) PP/MFC 80:20; (**b**) PP/MFC 60:40, magnification 250×.

**Figure 9 bioengineering-07-00103-f009:**
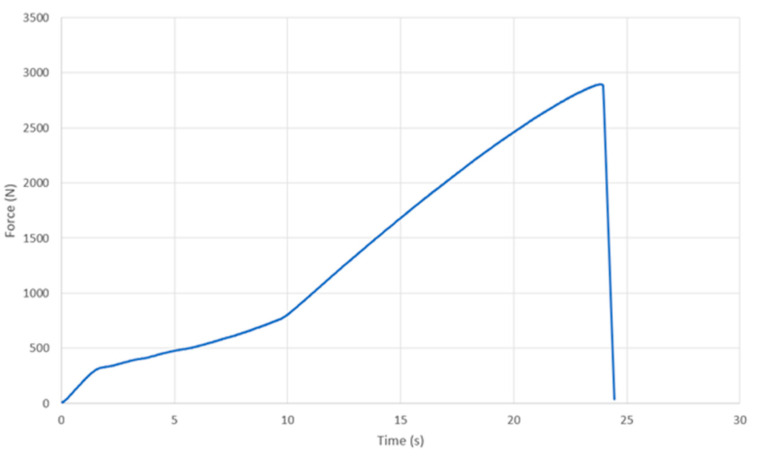
FDM-manufactured socket loading test curve in the sagittal plane. The testing method is illustrated in Figure 3.

**Figure 10 bioengineering-07-00103-f010:**
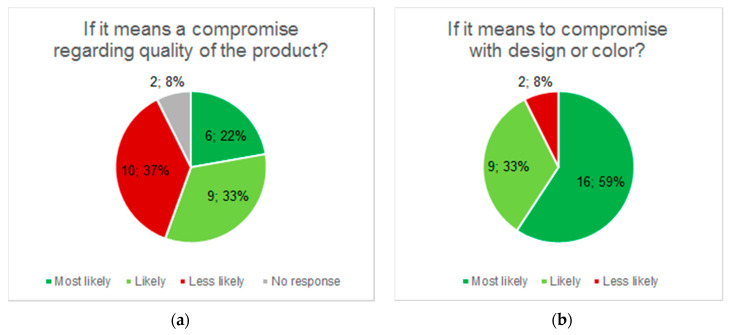
Consumers’ attitudes towards more sustainable materials in orthotic and prosthetic applications. (**a**) Quality aspect; (**b**) Design and color.

**Figure 11 bioengineering-07-00103-f011:**
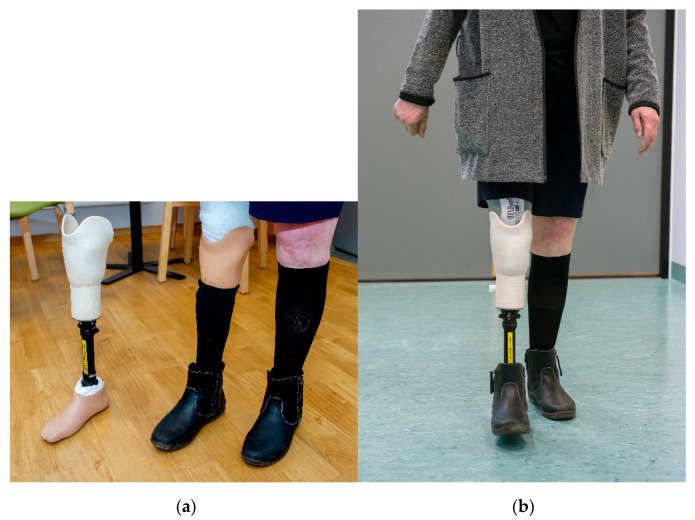
The demonstrator tested by a 70-year-old female with 52 years of experience using a transtibial prosthesis. (**a**) The patient wearing her conventional prosthesis, with the demonstrator to the left for comparison; (**b**) The patient walking with the demonstrator.

**Figure 12 bioengineering-07-00103-f012:**
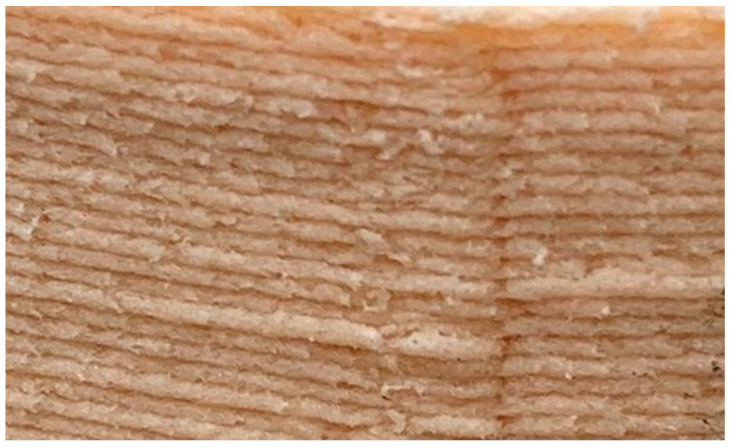
Illustration of defects (voids and pores) in an FDM structure.

**Figure 13 bioengineering-07-00103-f013:**
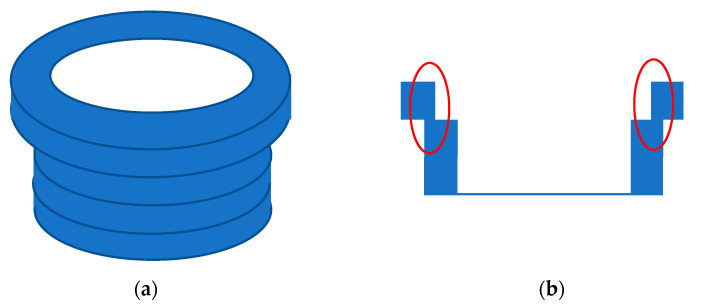
Illustration of reduced binding area between adjacent layers due to geometrical gradient. (**a**) The full structure; (**b**) The cross-section.

**Table 1 bioengineering-07-00103-t001:** Injection molding parameters.

Temperature Profile from Nozzle	200/200/195/190/180 °C
Injection pressure	Ramping down from 100 to 40 MPa
Injection speed	Ramping down from 20 to 3.8 cm^3^/s
Hold pressure	60 MPa
Hold time	8 s
Mold temperature	40 °C
Cooling time	20 s

**Table 2 bioengineering-07-00103-t002:** Mechanical properties of the composite injection-molded test specimens.

MFC Content (wt%)	$\mathrm{Tensile}\;\mathrm{Strength}\;\sigma_{m}\;(\mathrm{MPa})$	$\mathrm{Tensile}\;\mathrm{Modulus}\;E_{t}\;(\mathrm{MPa})$	$\mathrm{Elongation}\;\mathrm{at}\;\mathrm{Break}\;\varepsilon_{b}\;\left(\%\right)\;$	U_N_ [kJ/m^2^]
0	24.00 ± 0.81	1715 ± 97	>20	7.18 ± 0.40
20	32.10 ± 0.15	3112 ± 99	5.9 ± 0,7	3.79 ± 0.12
30	39.57 ± 0.24	3896 ± 57	4.5 ± 0,2	4.14 ± 0.14
40	40.81 ± 0,29	4525 ± 94	3.8 ± 0,2	3.41 ± 0.47

**Table 3 bioengineering-07-00103-t003:** Mechanical properties of the FDM-printed test specimens with the 20 wt% MFC content.

Sample	$\mathrm{Tensile}\;\mathrm{Strength}\;\sigma_{m}\;(\mathrm{MPa})$	$\mathrm{Tensile}\;\mathrm{Modulus}\;E_{t}\;(\mathrm{MPa})$	$\mathrm{Elongation}\;\mathrm{at}\;\mathrm{Break}\;\varepsilon_{b}\;\left(\%\right)$
Longitudinal print lines	21.96 ± 0.7	2100 ± 73	4.1 ± 1.3
Perpendicular print lines	16.11 ± 2.1	1885 ± 222	1.7 ± 0.4

**Table 4 bioengineering-07-00103-t004:** Test results of the FDM-manufactured sockets with the same material (PP and 20 wt% MFC), but in different printing directions.

Build Orientation	Force at Break (N)	Strength (MPa)
Transverse plane	1343 ± 82	6.03 ± 0.27
Coronal (frontal) plane	2385	10.7
Sagittal plane	2896	12.9

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
