# Peer review of "Additive Manufacturing of Prostheses Using Forest-Based Composites"

_bioengineering, 2020, doi:10.3390/bioengineering7030103_

Round 1
Reviewer 1 Report
- For my previous comment #3, the authors did not answer directly to my concern about cyclic mechanical testing. Do authors conduct cyclic testing or not? If not, authors have to add reasons and discussion into the manuscript instead of just describing in the comment response.
- Text for the response of comment #5 is different from the text in the manuscript. They have to be consistent.
Author Response
We thank you for your comments and suggestions. The manuscript has been revised accordingly.
- The following text is added in the manuscript as part of the limitations of this work. “Besides to the dynamic testing, cyclic mechanical testing is also important for prosthetic applications, which needs to be complemented in the future. “
- Fixed.

Reviewer 2 Report
The author has addressed my concerns, so it can be accepted for publication as it is now.
Author Response
Thanks!
This manuscript is a resubmission of an earlier submission. The following is a list of the peer review reports and author responses from that submission.
Round 1
Reviewer 1 Report
Publication of this manuscript is not recommended in the current format.
The authors should pay close attention to redefining the rationale to conduct this work, showing the shortcomings of the field.
The experimental design should accurately reflect the hypothesis and the methodology. Where necessary, more detail should be provided rather than self citations.
Mechanical and microscope analyses should be analyzed with respect to the hypothesis.
All inflated and over exaggerated claims should be removed or quantified.
The novelty and invocation is lacking because commercial blends are used. How would the authors justify a contribution to the field in this scenario?
The English language lacks style and sophistication. Proofreading and removal of errors is required throughout the manuscript.
graphics are poorly designed. More attention should be provided to follow standard, professional presentation.
Author Response
Dear reviewer,
We acknowledge your constructive and valuable questions, comments and suggestions. The manuscript has been extensively revised according the reviewer’s reports. English editing by professionals has also been made.
We hope you will find our responses (attached) satisfactory.

Reviewer 2 Report
Authors adopted micro fibrillated cellulose (MFC) fibres to reinforce polypropylene (PP) to manufacture prosthesis with fused deposition modelling (FDM) of 3D printing. The clinical trial of the prosthesis was conducted with successful outcomes in wearing experiences, appearance (colour), and acceptance towards the materials and the technique. However, there are many issues in the article that are not clearly described.
- Micro fibrillated cellulose, as a part from natural fibers, the fatal defects for the composites for application is that MFC holds strongly water and difficult to get a completely dried cellulose material in the matrix of PP, as described in Line 252. Therefore, it is necessary to test the wet mechanical properties to simulate the real mechanical property loss under wet condition.
- As for Figure.4 and Figure 5, error bars of the data should be added into the figure.
- Hydrophilic character of the fibres, in contrast to the hydrophobic nature of polypropylene always results in poor compatibility between PP and fibers and fibers tends to agglomeration and also poor mechanical properties. What strategies did the author adopt to avoid the effects mentioned above and realize even dispersion in Fig.5 and 6?
- The addition of fibres will increase the viscosity of the composites during the processing of the samples, especially in this paper the fibre contents as high as 20wt%. The dynamic viscosity of the composites should be added to evaluate the effects on the printing process.
- There are some relevant references missing, such as, Composites Science and Technology, 2019, 184, 107887; ACS Sustainable Chemistry & Engineering, 2019, 7, 18453-18462; Journal of Macromolecular Science, Part B Physics, 2011, 50, 907-921; Journal of Materials Science 2010, 45, 3520-3528, etc.
Author Response
Dear Sir/madam,
We acknowledge your constructive and valuable questions, comments and suggestions. The manuscript has been extensively revised according the reviewers’ reports. English editing by professionals has also been made.
We hope that you will find our responses to your comments satisfactory.

Reviewer 3 Report
This study presented additive manufacturing of a prostheses using composite materials of polypropylene reinforced by micro fibrillated cellulose fibres. Mechanical properties of composites and customer attitude towards the prostheses were investigated. Generally, this study is interesting and the manuscript is well prepared. However, there are some comments needed to be addressed to improve the manuscript.
- Since authors highlight that this study is a continuation of some published studies, they have to describe the relations of this study with the published ones.
- In introduction, as authors state, AM has been used to fabricate lower limb prostheses. However, literature about existing 3D printed prostheses for lower limb is missing. Also, authors need to clarify the advantages and novelty of this present prostheses over others.
- Durability of lower limb prostheses is important since it suffers cyclic loading daily while the patient is walking. Hence, cyclic mechanical testing for the material (or even the entire prostheses) is crucial besides tensile and compression tests.
- Figure 4, standard deviation is missing, and authors need to clarify how many samples are tested.
- “The tensile strength and modulus of the FDM test specimens built with longitudinal lines are about 2/3 of the values of the injection moulded shown in Table 2.” How do authors judge that the mechanical properties of 3D printed ones are sufficient? Comparison with mechanical properties with human bone from literature might be helpful.
- Some quantitative data from customer attitude towards the demonstrator would be more convincing.
Author Response
Dear Sir/madam,
We acknowledge your constructive and valuable questions, comments and suggestions. The manuscript has been extensively revised according the reviewers’ reports. English editing by professionals has also been made.
We hope that you will find our responses to your comments satisfactory.
Best regards.
